# Distributed genotyping and clustering of *Neisseria* strains reveal continual emergence of epidemic meningococcus over a century

Ling Zhong[1,2,9], Menghan Zhang[3,9], Libing Sun[4,9], Yu Yang[1], Bo Wang[3], Haibing Yang[3], Qiang Shen[3], Yu Xia[3], Jiarui Cui[3], Hui Hang[3], Yi Ren [5], Bo Pang[6], Xiangyu Deng [7], Yahui Zhan [3] ✉, Heng Li [1,2,8] ✉ & Zhemin Zhou [1,2,6] ✉

Core genome multilocus sequence typing (cgMLST) is commonly used to classify bacterial strains into different types, for taxonomical and epidemiological applications. However, cgMLST schemes require central databases for the nomenclature of new alleles and sequence types, which must be synchronized worldwide and involve increasingly intensive calculation and storage demands. Here, we describe a distributed cgMLST (dcgMLST) scheme that does not require a central database of allelic sequences and apply it to study evolutionary patterns of epidemic and endemic strains of the genus Neisseria. We classify 69,994 worldwide *Neisseria* strains into multi-level clusters that assign species, lineages, and local disease outbreaks. We divide *Neisseria meningitidis* into 168 endemic lineages and three epidemic lineages responsible for at least 9 epidemics in the past century. According to our analyses, the epidemic and endemic lineages experienced very different population dynamics in the past 100 years. Epidemic lineages repetitively emerged from endemic lineages, disseminated worldwide, and apparently disappeared rapidly afterward. We propose a stepwise model for the evolutionary trajectory of epidemic lineages in *Neisseria*, and expect that the development of similar dcgMLST schemes will facilitate epidemiological studies of other bacterial pathogens.

Since its first isolation in 1879, the population structure of *Neisseria* has been a subject of scientific interest[1], partially due to the challenge of phenotypically distinguishing the two pathogenic species, *Neisseria meningitidis* and *Neisseria gonorrhoeae*, that together cause ~90 million infections annually[2,3] from other commensal *Neisseria* species that are part of the normal nasal microbiome. Moreover, certain genetic lineages in these pathogens are of great epidemiological importance. For example, among the 13 serogroups in *N. meningitidis*, serogroup A (MenA) has been responsible for most of the major outbreaks in the last century, especially in developing countries in Asia and Africa[4]. Continual replacements of the epidemic MenA clones by other clones with different genetic backgrounds were evidenced every 10–15 years[5],

¹Pasteurien College, Suzhou Medical College, Soochow University, Suzhou 215123, China. ²Key Laboratory of Alkene-Carbon Fibers-Based Technology & Application for Detection of Major Infectious Diseases, Soochow University, Suzhou 215123, China. ³Suzhou Center for Disease Control and Prevention, Suzhou 215004, China. ⁴Department of Pathology, East District of Suzhou Municipal Hospital, Suzhou 215000, China. ⁵Iotabiome Biotechnology Inc, Suzhou 215000, China. ⁶National Key Laboratory of Intelligent Tracking and Forecasting for Infectious Diseases, National Institute for Communicable Disease Control and Prevention, Chinese Center for Disease Control and Prevention, Beijing, China. ⁷Center for Food Safety, University of Georgia, Griffin, GA, USA. ⁸Suzhou Key Laboratory of Pathogen Bioscience and Anti-infective Medicine, Soochow University, Suzhou 215123, China. ⁹These authors contributed equally: Ling Zhong, Menghan Zhang, Libing Sun. ✉e-mail: zhanyahui14@126.com; hli@suda.edu.cn; zmzhou@suda.edu.cn

according to the multilocus enzyme electrophoresis (MLEEs) or sequence types (MLSTs)[6]. After the significant drop of MenA infections by over 99.9% due to the introduction of specific vaccines[7], MenB and MenW have been responsible for increasing numbers of infections over the past decade[8]. This includes a new epidemic lineage in serogroup W135[9], which has recently been associated with multiple major outbreaks worldwide[10]. It remained unknown whether the rise of the W135 lineage was due to the artificial selection by the vaccine, or part of the continual replacements of epidemic clones. Furthermore, it is important to investigate the transmission pattern and population dynamics of the epidemics in the past, to better predict and control the epidemic lineages in the future.

Recently, core genome MLST (cgMLST) schemes have been established for many bacterial pathogens including *N. meningitidis*[11] and *N. gonorrhoeae*[12]. Based on 1422 and 1649 conserved genes in each species, cgMLSTs capture the majority of genetic variations in the core genome while maintaining scalability for analyzing >100 K strains[13], making it suitable for analyses spanning from disease outbreak surveillance[14] to species assignments[15]. However, their application in epidemiology was limited by three issues. First, the calculation and storage demands for cgMLST schemes have become increasingly intensive, because they require central databases[16] for the nomenclature of new alleles and STs, which must be synchronized worldwide[17]. Secondly, many genomes have never been uploaded into the central database due to proprietary and confidentiality concerns, which limited the cross-departmental collaboration in epidemiological investigations[18]. Finally, each bacterium has undergone a complicated phylogenetic history spanning a wide spectrum of genetic diversities[19], requiring a multi-level clustering scheme, i.e., HierCC[20], to represent natural bacterial populations.

In this work, to elucidate the population structure of the *Neisseria* genus, we show a distributed cgMLST (dcgMLST) scheme that does not require a central database of allelic sequences. The scheme is employed to divide 69,994 genomes into a hierarchical clustering scheme spanning genetic diversities from species (HC1050 + HC1130) to natural populations (HC760) and local disease outbreaks (HC10). We revise the current taxonomy in *Neisseria* and identify nine potential new species. We subdivide *N. meningitidis* into 171 lineages, including three epidemic lineages responsible for ≥9 epidemics or pandemics in the past 100 years. We reveal a relatively stable context of endemic

lineages for *N. meningitidis* in the long term, and the epidemic lineages emerged from it through a step-wise evolutionary trajectory. Furthermore, we demonstrate the use of HC10 for epidemiological investigation in five disease outbreaks. This novel scheme provides further insight into the taxonomy and evolutionary history of *Neisseria*, offering a valuable technique for identifying and tracking disease outbreaks.

## Results

### Distributed cgMLST scheme and the species tree based on a global dataset of 70,000 *Neisseria* genomes

To set up the new dcgMLST scheme, we established a global collection of genomic sequences for 69,994 *Neisseria* strains (Supplementary Data 1), consisting of 4411 assembled genomes from GenBank, 65,434 genomes assembled based on short reads deposited in the NCBI SRA database, and 149 novel genomes for *N. meningitidis* isolates from Suzhou, China in 1975–2021. The final collection consisted of strains from the United Kingdom ($n = 14,536$), the United States ($n = 14,461$), Norway ($n = 2511$), China ($n = 413$), and 90 other countries (Table 1). Although the majority of the strains were isolated in the past decade, there were 333 strains isolated during 1915–1979, and 1913 strains from the 1980s and the 1990s, resulting in the most comprehensive collection so far for the investigation of historic epidemics (Table 1).

We selected a set of 7630 representative sequences that encompassed most of the genetic diversity in the global collection and used them as the basis to establish a cgMLST scheme consisting of 1149 core genes that were shared by ≥95% of the representative genomes, following the same procedure outlined previously (see the Method for details). Furthermore, we applied the dcgMLST scheme to the global collection and found that 1146 of the core genes were still shared by ≥95% of the 69,994 strains, with no signal of collisions for the MD5 values. In particular, we invented an algorithm that designated each core gene allele as the MD5 hash value of the sequence of the gene, and thus enabled distributed cgMLST (dcgMLST) nomenclature without a central database (Fig. 1). Furthermore, we assigned each the dcgMLST allelic profiles using pHierCC into a set of 1149 hierarchical clustering levels, from HC0, namely no allelic differences, to HC1148 where genomes that differed each other by all but one alleles were grouped. Some of these clusters were found to be biologically meaningful since they were genetically separated from the others according to the Silhouette scores (Supplementary Fig. 1) and were consistent with species or populations found by other techniques. These included many clusters in the HC1130/HC1050 levels that resembled *Neisseria* species, HC760 clusters that resembled natural populations or clonal complexes (CCs) by 7-gene MLST, HC300/HC400 levels that represented clades within the populations, and HC10 clusters for genetically extremely closely related strains from the same disease outbreaks. To facilitate the application of the dcgMLST scheme, we implemented an automatic pipeline, DTy[21] (https://doi.org/10.5281/zenodo.8396234), which automatically genotypes a *Neisseria* genome and assigns it into multi-level HC clusters and species after comparing them with the representative genomes. Additionally, we built a species tree for *Neisseria* by integrating the 1149 core gene trees estimated from the 7430 representative genomes using a divide-and-conquer algorithm, as previously described[19] (see the Method for details).

### Revised species designations based on clustering of dcgMLST profiles

The taxonomy of *Neisseria* was known to be erroneous and incomplete[22]. To revise the species designations, we sub-divided the representative genomes into single-linkage clusters of 95% average nucleotide identities (ANI95%), which have been widely used as a proxy to bacterial species in many bacteria[23]. The ANI95% clusters in *Neisseria*, however, were inconsistent with species. For example, ANI95% inaccurately grouped *N. meningitidis* and *N. gonorrhoeae* as the same

**Table 1 | The number of isolates enrolled in this study**

|  | No. of isolates |
|---|---|
| **Sources** | |
| NCBI SRA | 65,434 |
| NCBI Genbank | 4411 |
| This project | 149 |
| **Countries** | |
| United Kingdom | 14,536 |
| United States | 14,461 |
| Norway | 2511 |
| Others | 38,337 |
| **Isolation years** | |
| 1915–1979 | 333 |
| 1980–1999 | 1913 |
| After 2000 | 42,839 |
| **Clonal complexes (CCs)** | |
| ST-1 complex | 537 |
| ST-5 complex | 785 |
| ST-11 complex | 6326 |
| Other CCs | 25,161 |

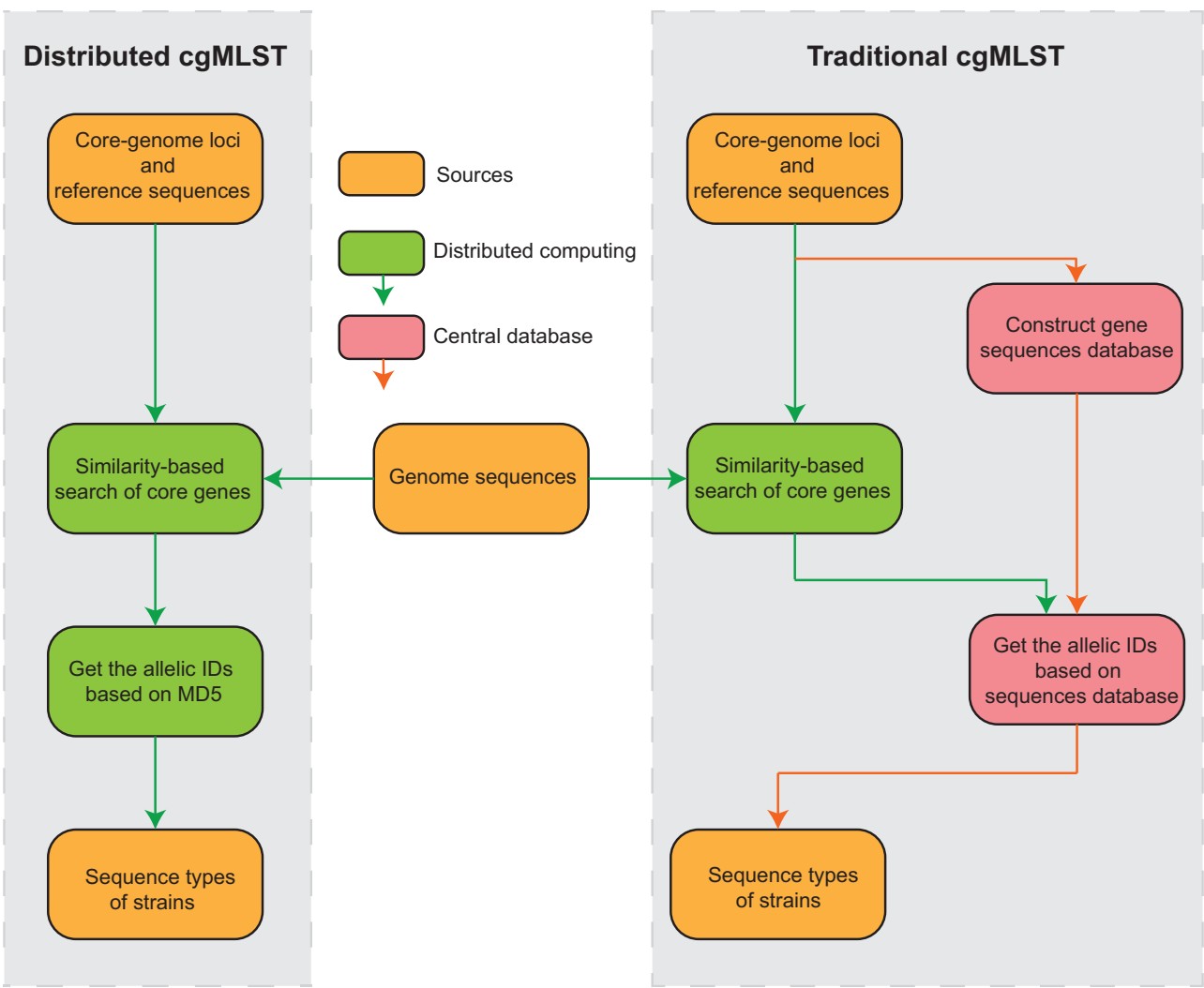

**Fig. 1 | Comparison between distributed cgMLST and traditional cgMLST.** Compared to traditional cgMLST, distributed cgMLST eliminates the dependence on a centralized database, thus improving the comprehensiveness, generality, and accuracy of genome sequence typing.

species. It also yielded only one large cluster for all genomes in the *N. polysaccharea* polyphyletic group, which was recently subdivided into at least 12 species in a recent study[15] (Supplementary Fig. 2a). We attributed the inconsistency between the ANI95% clusters and species to the high similarity between these species. Here we identified in the species tree (Fig. 2a) the presence of a species complex, named NM complex thereafter, which consisted of many human-specific cocci of *N. meningitidis, N. gonorrhoeae, N. lactamica, N. cinerea*, and the *N. polysaccharea* polyphyletic group. The inter-species ANIs in the NM complex were 91-96%, overlapping with the 94-100% ANI within each species (Supplementary Fig. 2b). Therefore, we decided to curate the species assignments in the NM complex manually following the traditional taxonomy, and only revise other species using the ANI 95% clusters (Supplementary Fig. 2b).

Here, we identified two HC levels, HC1050 and HC1130, that form clusters similar to species within and outside of the NM complex, respectively. All species outside the NM complex exhibited one-to-one correlation with the HC1130 clusters, except for *N. subflava* and *N. mucosa* which each corresponded to multiple HC1130 clusters (Supplementary Table 1). The HC1130 clusters in these two species corresponded to subspecies that were previously described as species and only merged recently. HC1130 assigned almost all genomes in the NM complex into one cluster of HC1130_1, which could be subdivided into multiple HC1050 clusters, each corresponding to one species with

high accuracies (ARI: 0.99). As such, HC1130 and HC1050 clusters offered an efficient alternative to both the traditional taxonomy and the ANI that allowed rapid assignments of strains into species based on the genomic sequences (Fig. 2b). Notably, we also identified 6 HC1050 clusters and 4 HC1130 clusters that were not associated with any existing species, leaving them potential new species to be investigated in greater detail.

## Long-term persistence of *Neisseria* population revealed by HC760

The HC760 clusters were the most cohesive according to their greatest average Silhouette score (Supplementary Fig. 1), and were similar to the CCs with little discrepancies (ARI: 0.97). We visually investigated the discrepancies that involved more than one strain and sub-divided them into three categories of (a) seven "merges", in which two CCs were assigned as the same HC760, (b) four "splits" where one CC was divided into two HC760s, and (c) three "conflicts", in which multiple CCs and multiple HC760s were intermixed (Supplementary Table 2). Comparing the discrepant clusters with the species tree, we found that HC760s outperformed CCs since the latter were polyphyletic in six cases (Supplementary Fig. 3). There were a total of 570 HC760s in the whole genus, including 171 HC760s in *N. meningitidis* and 60 in *N. lactamica*. Of note, the whole *N. gonorrohoeae* consisted of only one HC760_1, highlighting its lower genetic variations than other human-specific *Neisseria* (Fig. 2b).

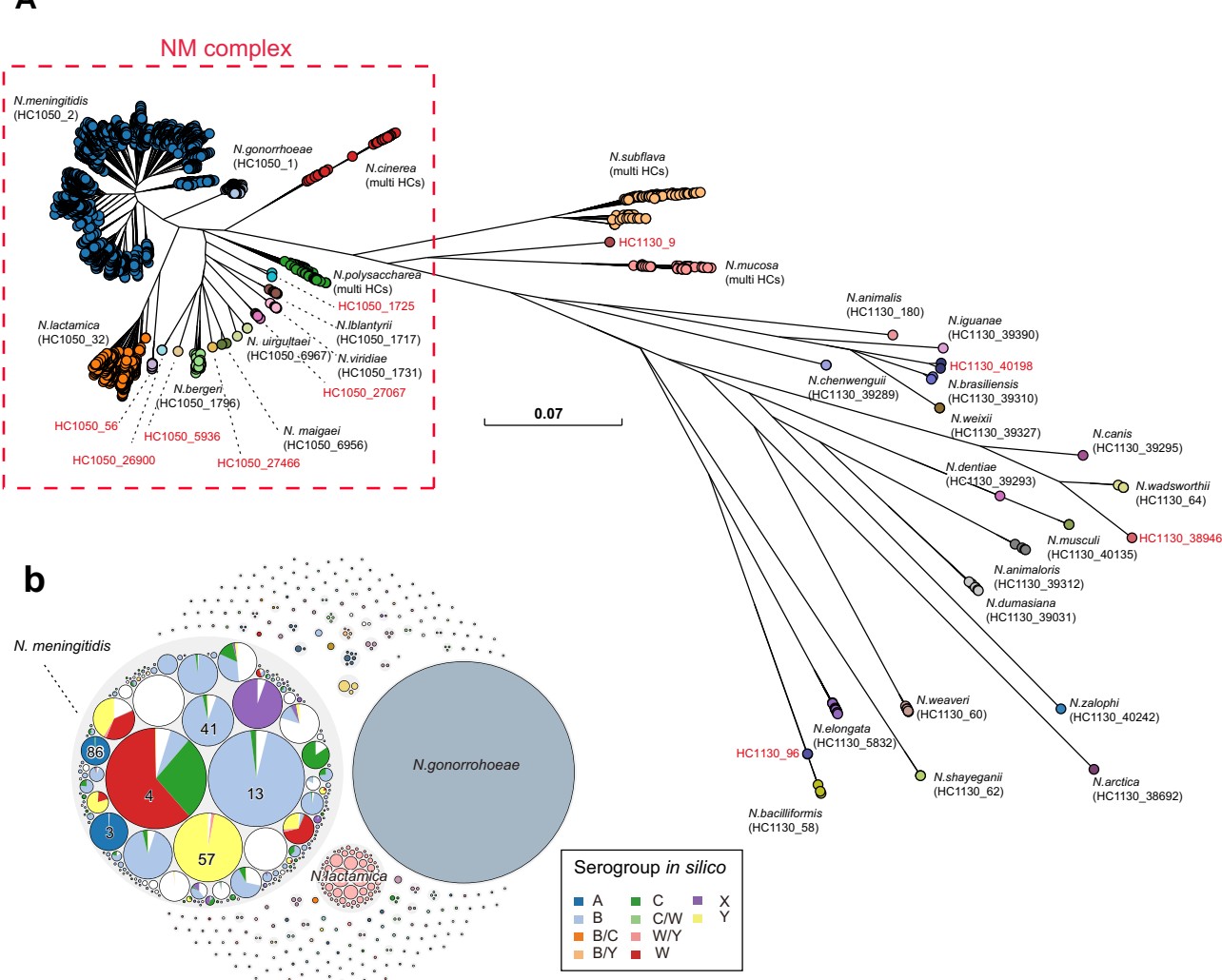

**Fig. 2 | Species tree and population structure of the *Neisseria* genus.** a Maximum-likelihood tree based on representative strains of the *Neisseria* genus. The red dashed border indicates the NM complex. **b** Hierarchical bubble plot for the population structure of the *Neisseria* genus. Large circles colored in gray presented *Neisseria* species and the colored circles inside presented HC760 lineages, which were color-coded by predicted serogroups (in *N. meningitidis*) or species. Source data are provided as a Source Data file.

Moreover, both HC760s and CCs correlated with the predicted pathogenic serogroups in *N. meningitidis* poorly (ARI value -0.4). Serogroups in *N. meningitidis* were known to be highly variable and underwent frequent horizontal gene transfers. The only exception was MenA, which was mostly associated with only two HC760s (HC760_3 and HC760_86) from a single monophyletic cluster in the species tree. Strains from these two HC760s accounted for ~45% of strains before the 1980s and were associated with most of the major disease outbreaks in the last century. The relative frequencies of these two HC760s dropped to only 3% in the past 40 years, which was likely associated with the application of MenA vaccines in epidemic regions (Fig. 3a). Meanwhile, we witnessed an increase of MenB and MenY strains in Europe (31–72%), MenW and MenC in Africa (<1% to 64%), and MenB, MenC, and MenW in Asia (20–65%) after 2000 (Fig. 3b).

### Evolutionary trajectories of the MenA HC760_3 lineage
To investigate the evolutionary history of the MenA isolates, we genomic sequenced 108 frozen stocks collected between 1975 and 1980 from the Suzhou Center for Disease Control and Prevention (CDC), and found that 79 (73%) were in serogroup A. These MenA strains were from only two clusters of HC760_3 (one strain) and

HC760_86 (78 strains). Together with publicly available genomes, we reconstructed the phylogenies and date origins for both MenA HC760s.

According to the phylogeny (Fig. 4a), HC760_3 originated in 1904 (CI 95%: 1898–1910) and subsequently diverged into two primary branches that could be designed as HC300 clades of 3 and 74. Thanks to the high consistency between clusters defined by cgMLST, MLST, and MLEE, we were able to associate HC760_3 with >5 epidemics worldwide in the 1930s, 1960s, 1980s, and 1990s[24]. The HC300_3 consisted of mainly ST4 in legacy MLST and lineage IV in MLEE. The two most diverged strains in HC300_3 were both from the US in the 1930s. These strains, although rare nowadays, were responsible for successive epidemics in Western countries before WW-II[25]. After the war, this clade was mostly isolated in West Africa with the time of most recent common ancestor (tMRCA) in 1950 (CI95% 1945–1955), resulting in massive disease outbreaks there in the 1960s and 1980s (Fig. 4b)[26].

HC300_74 was generally consistent with STs 5 and 7 in legacy MLST and lineage III in MLEE. This clade was likely present in China before 1908 (CI95% 1900–1916) and later disseminated globally. A previous study[24] subdivided strains in lineage III into nine so-called

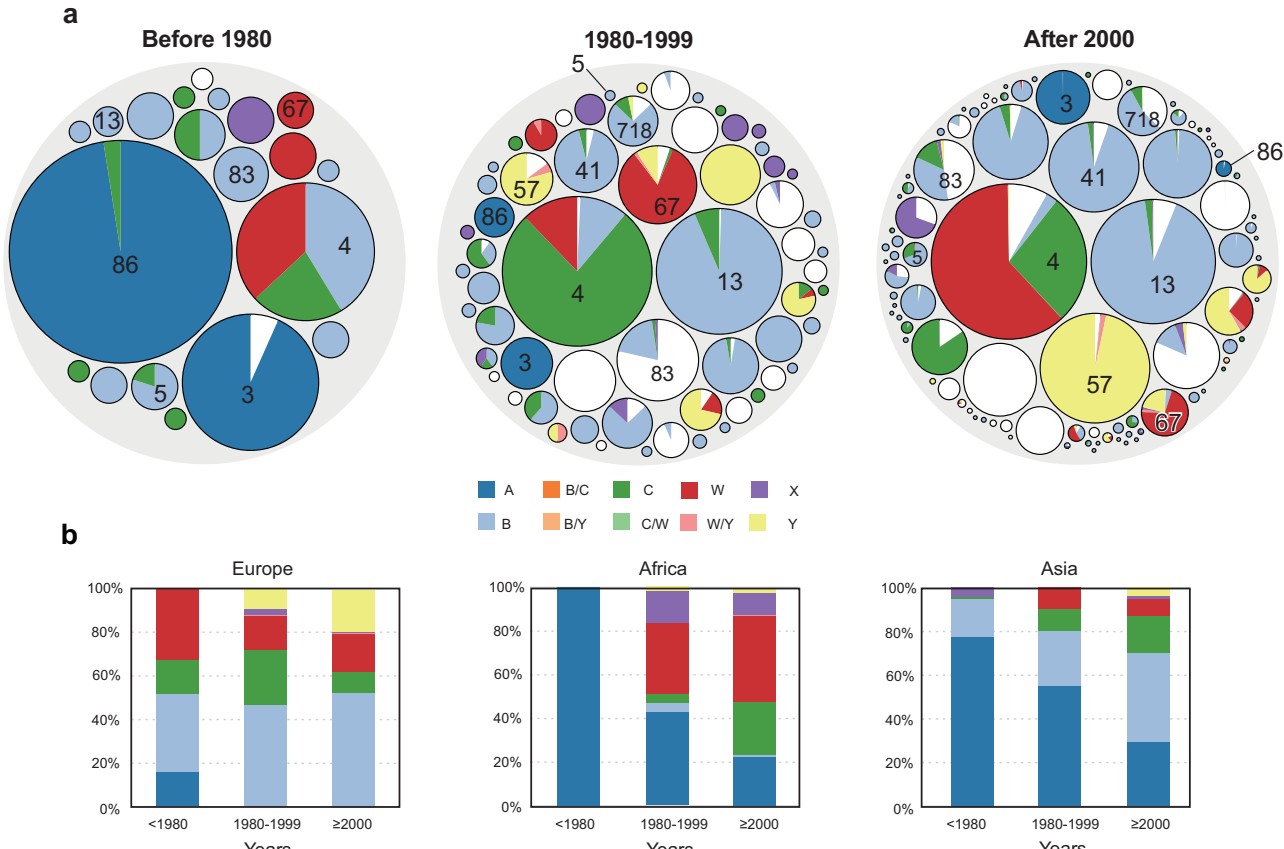

**Fig. 3 | Population changes of *N. meningitidis* with different colors represent different serogroups. a** Population structure of *N. meningitidis* in three different periods. Each small colored circle represents an HC760 lineage, and the number inside the circle is the ID of that HC760. **b** A stacked bar chart showing the percentage of pathogenic serogroups in Europe, Africa, and Asia across the three time periods. Source data are provided as a Source Data file.

"genoclouds" based on Restriction Fragment Length Polymorphism (RFLP) and sequence analyses of 6 genes, tracking its international transmission for the first time. To compare our results with these historical records, we retrieved five of these genoclouds based on the genomic data (Fig. 4a). Genoclouds 1 (G1) and 4 (G4) were only found in China. In particular, G1 belongs to the so-called 1st pandemic and was associated with the largest epidemic in China in the 1960s[25]. G3 was transmitted into Brazil in 1971 (CI95% 1900–1916), resulting in outbreaks between 1973 and 1976[25]. G5 was introduced into Africa in 1984 (CI95% 1982–1986), three years before its first report in the Mecca outbreak in 1987[27]. This genocloud caused the 2nd pandemic there, including the largest outbreak of meningitidis so far in the African meningitis belt in 1996[27]. Finally, G8 also spread out of China in the late 1980s into Bangladesh, the United States, and the United Kingdom, and arrived in Africa in 1993 (CI95% 1990–1996). Different from its predecessors, G8 stayed epidemic internationally for >20 years. By the 2000s, it formed at least five clones (genocloud clones; GCs) that were each associated with different countries. GC8.1 remained in China and had at least one strain transformed to serogroup MenX before 2014. GC8.2 stayed in Bangladesh but was also transmitted into the UK in 2010. GCs 8.3 to 8.5 circulated in Burkina Faso, Chad, and Ghana, respectively, and each transmitted into other countries in the African meningitidis belt. Meanwhile, we evidenced an expansion of effective population size for HC760_3 between 1980 and 2010, likely associated with the global dissemination of G5 and G8 (Fig. 4b).

### The evolutionary trajectories of HC760_86 MenA
Originating in 1891, HC760_86 was more divergent than HC760_3 and could be subdivided along the phylogeny into six HC300

clades (Fig. 4a). Two clades (HC300_86 and HC300_31213) were predominant, accounting for 88% of all HC760_86 strains and ≥3 epidemics in the 1950s, 1960s, and 1990s[28]. HC300_86 contained mostly ST1 in legacy MLST or lineages I and II in MLEE and had its MRCA in Germany in 1938 (CI95% 1926-1950). The clade was later transmitted to Algeria, and from there, spread into other countries in Africa and Asia, causing major disease outbreaks in the 1960s and 1970s[26]. This clade was also transmitted into South Africa in 1995 (CI95% 1990–1998), either directly from Algeria, or via Australia, and resulted in a national epidemic during 2001-2003 (Fig. 4b)[28].

Meanwhile, HC300_31213, the other major clade of HC760_86, stayed endemically in China for over 20 years since its tMRCA in 1960[7], accounting for 3/4 of the historical strains from Suzhou, China. This clade was also engaged in the disease epidemic in China in the 1960s, but has not been transmitted into other regions (Fig. 4b)[7]. Instead of forming province-specific clusters in the phylogeny like those at the country level, The Suzhou isolates intermixed with strains from other Chinese provinces in the 1970s, indicating a high level of inter-provincial transmissions. The effective population size of the HC760_86 lineage reached its peak between the 1960s and the 1970s and has been declining for more than 40 years since, in contrast with the population expansion of HC760_3 at the same time (Fig. 4c).

### The evolutionary trajectories of HC760_4, the new global epidemic lineage
A maximum-likelihood phylogeny tree was also built for HC760_4 based on 162,586 SNPs in the core genome of 3,975 global strains (Fig. 5a). We found that the five clades along the tree were perfectly

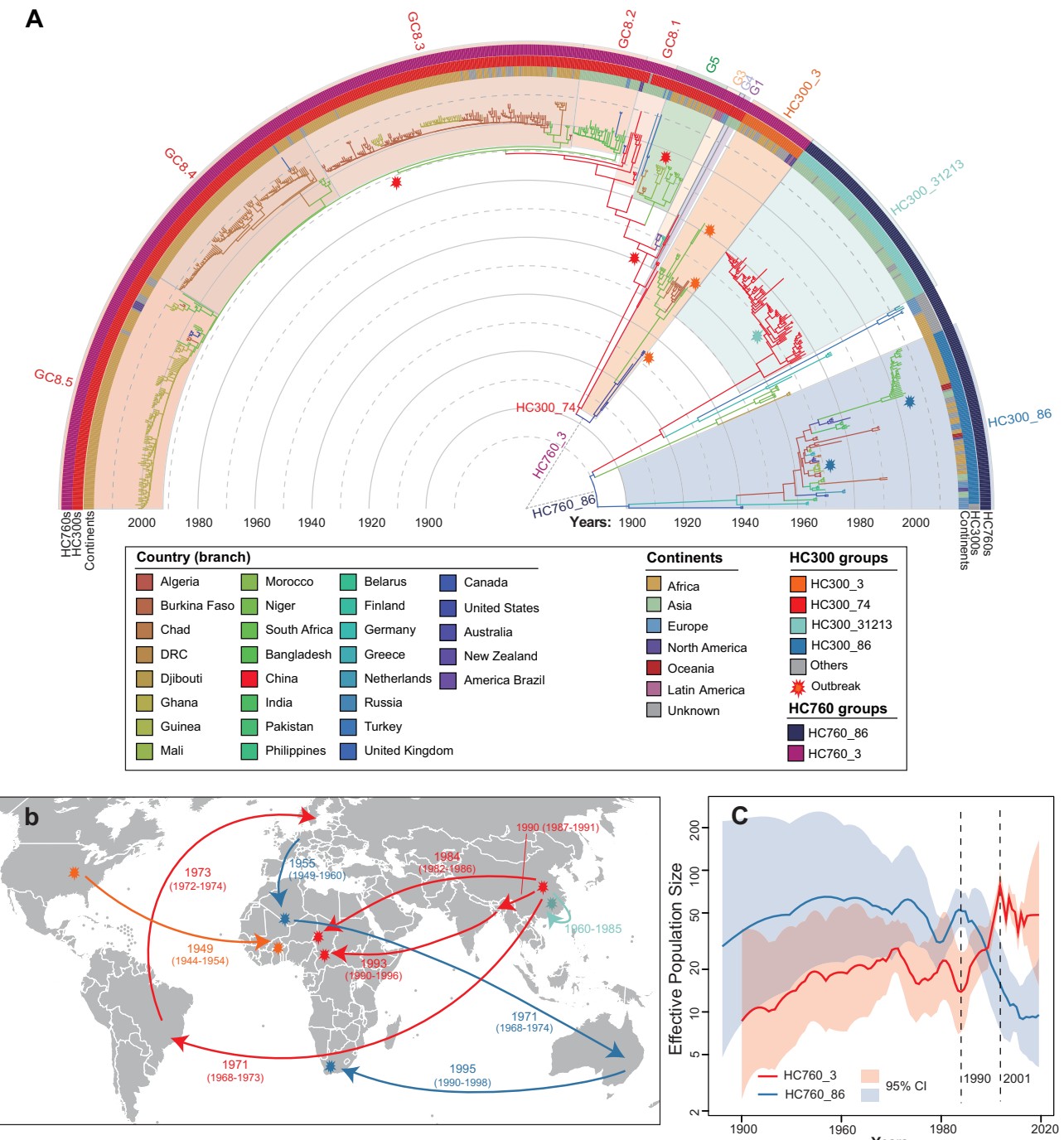

**Fig. 4 | Phylogeny and transmission of the *N. meningitidis* serogroup A (MenA) strains. a** Spatiotemporal phylogenetic tree of the MenA strains. The colors of the branches presented the predicted country by TreeTime. The arcs outside showed HC300 clades and isolated continents for the strains. The colored shadows along the tree showed the HC300 clades, genocloud ids (G1–G5), or clusters in genocloud 8 (GC8.1–GC 8.5). Stars highlighted pathogens associated with major disease outbreaks (also in part B). **b** Global transmissions and disease outbreaks of MenA. The number near each line showed the inferred year of transmission by BactDating. The colors of the lines indicated the associated clades, genoclouds, or clusters (as in part A). The world map was modified from the map hosted in the d3.js (https://d3js. org/). **c** Dynamics of effective population sizes (Y-axis) by years (X-axis) for the HC760_3 (red) and HC760_86 (blue) lineages. The 95% confidence intervals were shown as shadows. Source data are provided as a Source Data file.

resembled by HC400s, including HC400_4 (3898 strains), 45 (66), 120 (1), 1488 (1), and 31114 (9). HC760_4 contained two major CCs of ST8 complex (HC400_45) and ST11 complex (HC400_4), both of which were hypervirulent and responsible for many meningitidis outbreaks after 2000[29]. Besides, we also found a cluster of 9 Chinese strains (HC400_31114) that was closely related to the ST11 complex and also associated with disease outbreaks in China (see below).

The HC400_4 clade had a time of MRCA in 1891 (CI95% 1884-1897), approximately 100 years before it became a recognized clone, and experienced continuous population expansions after the 1960s. Following the SNP tree, we sub-divided the HC400_4 clade into six clusters that each disseminated in different countries/regions, and designated them following the associated serogroups as C-I, C-II, W-0, W-I, W-II, and W-III (Fig. 5a). Clusters C-I and C-II mainly circulated in

Europe and the US, with a few transmissions into New Zealand in 2000 and Vietnam in 2015 (Fig. 5c). One sub-branch in C-II converted its serogroup into W135 before 1962, leading to the emergence of Cluster W-0, which was the ancestor of other MenW clusters. W-0 stayed mainly in Europe except for one transmission into South Africa in 1972. The other MenW clusters were transmitted into other regions more often. Cluster W-I was transmitted into Africa multiple times between 1993 and 1999, and became epidemic in Africa in the early 21st century. It was the pathogen responsible for the infamous Hajj outbreak in 2000. Cluster W-III was also transmitted into Africa after 1992 and into China in 2008. All Chinese W135 strains, including the seven strains from Suzhou, were from Cluster W-III. Finally, different from other clusters that all originated in Europe, Cluster W-II originated in South America in 1986 and transmitted into the US, Europe, and Australia in the early 2000s, responsible for the majority of W135 strains there (Fig. 5c).

### Long-term persistence of endemic lineages globally and locally

Apart from the rapid rise and fall of the epidemic lineages in the past 40 years, we found that almost all other lineages nowadays had already been isolated before 2000 (Fig. 3a), with only little fluctuation of their frequencies. Furthermore, most of these endemic lineages have also been reported before 1980 despite the dominance of MenA strains at that time. To further examine the long-term persistence of endemic lineages, we compared the Suzhou isolates from the 1970s and nowadays. Interestingly, five HC760s (HC760_2, 4, 5, 153, and 26,061) co-existed in both periods, accounting for 61% (17/28) and 75% (24/32) of the endemic strains in both periods, respectively. Six HC760s were only found in one of both periods. Each of these HC760s contained only 1–2 strains except for HC760_19454 which likely represented a cluster of 5 isolates from a disease outbreak in 1975.

### Reconstructing transmission chains of disease outbreaks

To demonstrate the use of dcgMLST for tracking disease outbreaks, we investigated five sets of Suzhou strains that had epidemiological records. Each set of strains consisted of an isolate from the patient with invasive meningitidis and additional isolates from close contacts (Fig. 6). Intriguingly, four of these five patients was associated with strains from HC760_4, including three from HC400_4 (CC11) and one from HC400_31114. The remaining patient was infected by HC760_5 (CC4821), a prominent cause of invasive meningitidis infections in China[30]. By using the HC10 groups as a proxy for identifying genetically closely related strains, we found that the genomic data and the epidemiological data only completely agreed in one case, where the strain (S186) from the patient (daughter) shared the same HC10_70135 with the strain (S187) from her mother. Besides, genomic data partially agreed with the epidemiological records in three cases, where some close contacts carried strains from the same HC10 clusters as the strain from the patient. For example, in one case, the patient's strain (S48) shared the same HC10_70151 with strains from his/her relatives, but strains from five co-workers were genetically very different, and probably not part of the transmission chain. Finally, there was one case that the strain S4 from the patient was completely different from strain S59 from the close contact, even though the latter was also diagnosed with meningitidis later. This could either be infections from unrelated strains, or from transmitted strains that haven't been characterized in the culture.

## Discussion

First proposed by ref. 31, at least 50 cgMLST schemes for different species and/or genera have been established in the past decade, covering more than 700,000 bacterial genomes[17]. The technique has been successful and is used as a standard tool for monitoring disease outbreaks and tracking transmission chains. However, its further application has been hampered by the complexity mentioned earlier (see Introduction). Here, we proposed an extension to cgMLST that allowed

decentralized allele callings. Using the MD5 hash proposed here, each allelic sequence in the core gene was assigned a unique integer, and different allelic sequences were almost guaranteed to have different numbers (expecting one collision in 9 trillion). Thus, the genome was presented by a set of alleles that were fully compatible with all existing tools for cgMLST, but no longer required a centralized nomenclature database (Fig. 1).

The new dcgMLST scheme facilitates data exchange in two ways. First, the only data required for synchronization between data centers are the genotypes of the strains, which are merely 1000s integers for each strain or up to several GBs of data for 1 million strains. For example, the allelic profiles of ~70,000 Neisseria genomes occupied 67.2MB of storage space and could be easily shared with the DTy pipeline in public storage spaces. As a comparison, the Clostridioides cgMLST in EnteroBase currently hosts 29,085 genomes, ~40% as large as the Neisseria genomes here, and has registered 463 K alleles that account for ~600 MB of storage spaces. The dcgMLST also replaced the allelic comparison process, which scales linearly with the size of the database, to MD5 hash of constant time complexity. Additionally, MD5 hash is an irreversible encryption, and the MD5 values cannot be reversely decoded into sequences. Therefore, strains characterized by dcgMLST have a minimum risk of leaking their genomic sequences. Finally, we decided not to use the set of core genes hosted in pubMLST, because our scheme covered strains from the whole Neisseria genus, which was different from the species-specific schemes that were currently in use[11,12].

There are, of course, some limitations for dcgMLST. Multi-locus sequence analysis (MLSA), a detailed phylogenetic analysis based on the allelic sequences of genes in the MLST schemes[32], will hardly apply to records in the dcgMLST scheme due to a lack of sequence information. Also, while the allelic designations have been de-centralized, there is still a need for public databases for sharing allelic profiles and metadata of bacterial strains for biological interpretation of the results.

Based on the new dcgMLST scheme, we built a tree comprehensively describing the genetic diversity of the Neisseria genus, representing 7630 sequences selected from ~70,000 genomes. We managed to recover most of the species that were previously described based on genomic data[15,22] and identified a new NM species complex where ANI95% was not working properly for species assignment. Notably, Diallo et al. [15], managed to separate NM complex into meaningful ANI95% clusters. This is because they included only <10% of the strains in the investigation. Many ANI95% clusters they identified were merged into one due to the intermediate strains that were only included in this study. We showed here that the HC levels may be more appropriate for genome-based species assignment. Based on two HC levels of HC1050 and HC1130, any genome could be assigned to species accurately and efficiently. Furthermore, HierCC also automatically found 10 clusters in Neisseria that were not associated with any existing species, suggesting them as new species to be characterized. Similar approaches have also been employed for four genera hosted in EnteroBase[17], resulting in the observation of dozens of new species there as well.

Here, we proposed using HC760 as a replacement for CC from the legacy MLST because the HC760 clusters were more consistent with the phylogeny than the CCs. We showed that HC760 clusters likely represented natural populations in N. meningitidis, and could be employed to track the long-term evolutionary history of epidemic lineages over hundreds of years. Applying HC760 clusters to other Neisseria species would potentially broaden our understanding of these microbes, since there is currently no available designation for populations in these species at all. Interestingly, all genomes in N. gonorrhoeae were from only one HC760 cluster. This is possibly associated with its relatively recent tMRCA in the early 19th century[33].

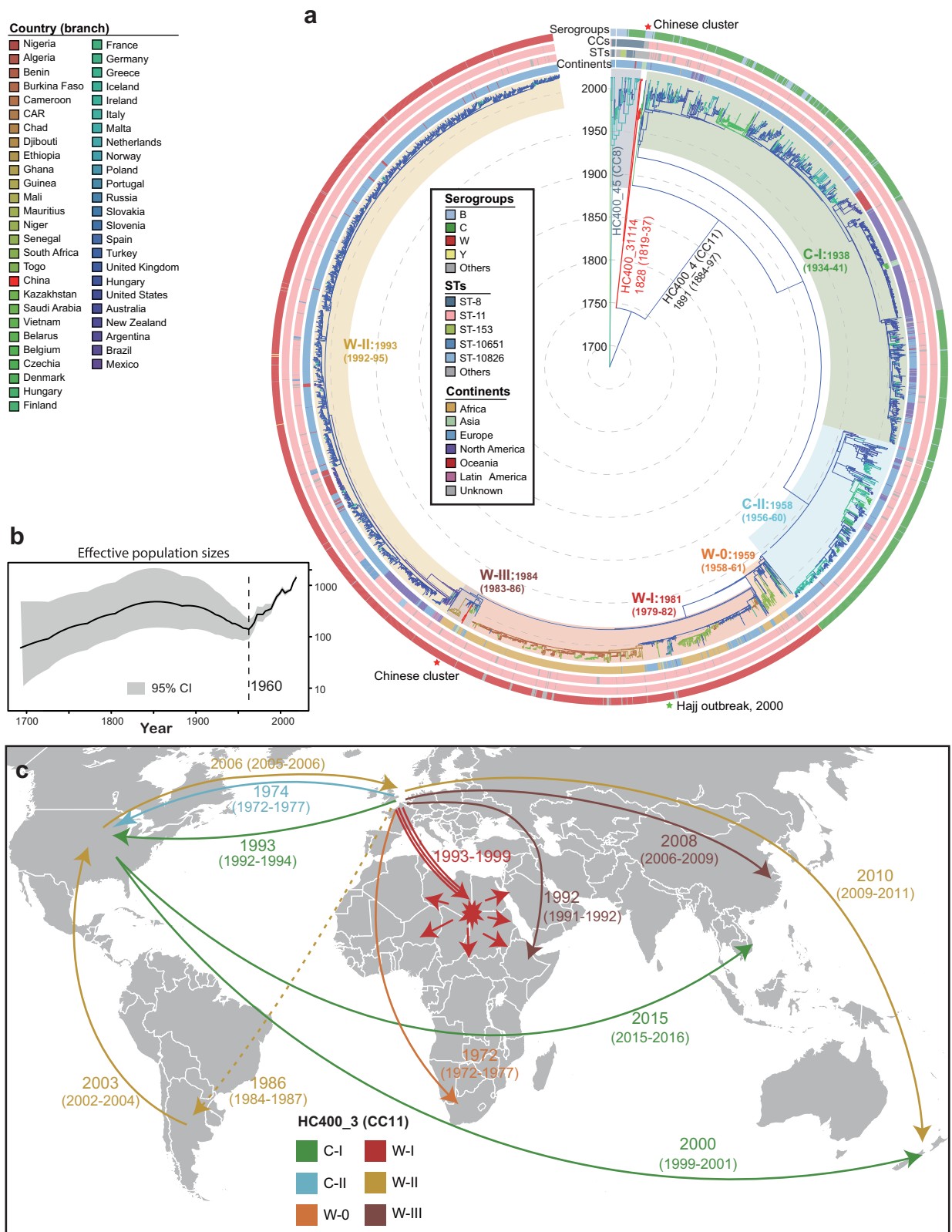

**Fig. 5 | Global transmission of the HC760_4 lineage. a** Spatiotemporal phylogenetic tree of HC760_4 group. The colors of the branches represent the predicted original country based on TreeTime (left Key). The circular bars surrounding the tree showed (from outer to inner) serogroups, STs, CCs, and isolation continents of the strains (inset Keys). **b** The plot of effective population sizes (Y-axis) by years (X-axis) for the HC760_4 lineage. The 95% confidence intervals were shown as shadows. **c** Cartoon of the major global transmission events and disease outbreaks. The numbers near each line showed the inferred year of transmission by Bact-Dating. The colors of the lines indicated the associated clades as in the Key. The world map was modified from the map hosted in the d3.js (https://d3js.org/). Source data are provided as a Source Data file.

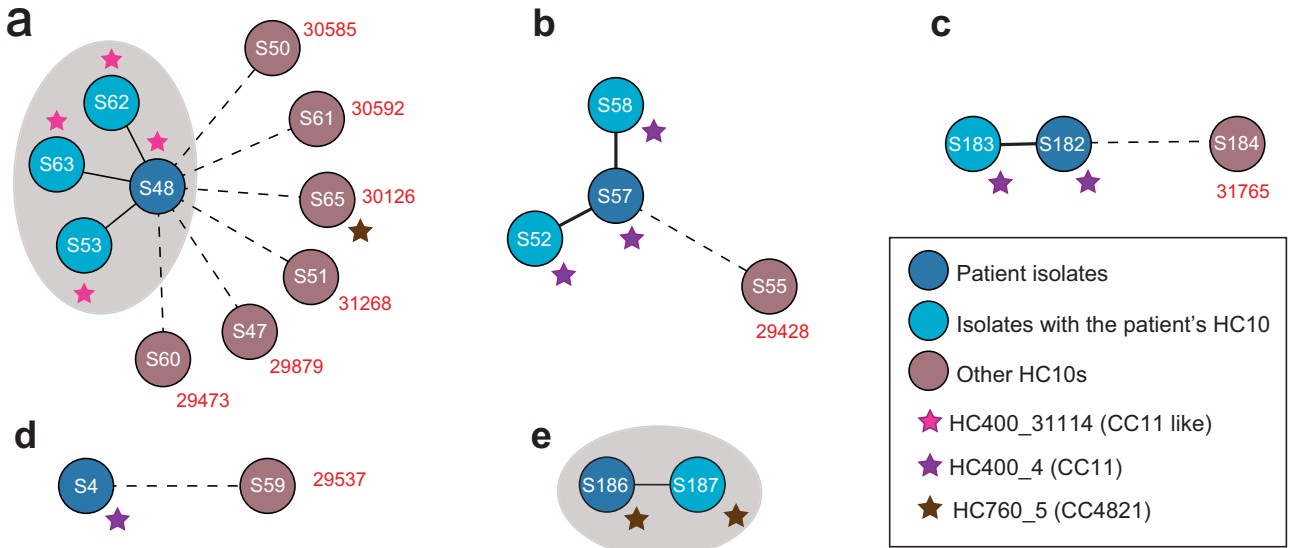

**Fig. 6 | Evaluation of potential transmissions in five disease outbreaks using HC10 clusters. a–e** The central node filled with dark blue in each plot showed the isolate from the patient. Nodes (light blue) linked to the patient with solid lines showed isolates from the close contacts that shared the same HC10 cluster with the patient, and the gray nodes showed isolates from the close contacts but in a very different HC10 cluster. The red numbers close to the gray nodes showed the number of core SNP differences in the patient. The gray shadow in parts a and e grouped people from the same families together. The stars nearby some nodes highlighted epidemic lineages according to the key. Source data are provided as a Source Data file.

Here we visualized the dynamic population changes in *N. meningitidis* over time. Comparing strains before 1980 and after 2000, the relative frequencies of epidemic lineage fluctuated significantly. The historical MenA lineages decreased from 63% to 3%, probably associated with the extensive use of vaccines, while the HC400_4 (ST11 complex) lineage increased from 12% to 22%, becoming the predominant lineage. Meanwhile, most of the local endemic lineages, i.e., HC760_5 (ST4821 complex) in China and HC760_90 (ST174 complex) in the UK, seem to have been relatively stable over time. Similar population dynamics have also been reported 50 years ago using serogroups. As reported, although MenA was responsible for multiple epidemics, MenB and MenC remained predominant in the non-epidemic years in the US[34] between the 1920s and the 1970s.

Thanks to the good association between dcgMLST, MLST, and MLEE, we managed to retrieve the continuous switches of the epidemic lineages by integrating historical records and genomic data. We found that most meningitidis outbreaks in the African meningitis belt were associated with MenA ST1 in the 1960s, ST4 in the 1970s and 1980s, shifted back to ST1 again in the 1980s, and finally replaced by ST7 G8 from China in the 1990s. A "genocloud" model was previously proposed to explain the transition of epidemic lineage during pandemics as a result of competition between populations[24] This model, however, assumed the everlasting of epidemics which is unrealistic. In contrast, the "transient Darwinian selection" model[35] described the extinction of epidemic lineages by long-term purifying selection with the endemic lineages. We, therefore combined both models into a 5-stage evolutionary scenario, as described below.

As mentioned above, the population structure of endemic lineages in *N. meningitidis* was relatively stable over the long term (Fig. 7a). Some endemic lineages would tend to expand their population sizes due to short-term positive selection or by genetic drift (Fig. 7b). For example, the MenA HC760_3 (ST5 & 7) was endemic in China for >60 years, and HC760_4 (ST11) lineage was endemic in Europe for >100 years, both experienced continuous population expansions during the endemic phase. These emerging lineages also diverged into multiple clades, some of which transmitted into other countries/regions (Fig. 7c). For example, HC760_3 transmitted into Bazil and HC760_11 transmitted into New Zealand and South Africa. However, most of these early transmissions would not lead to long-lasting disease outbreaks, possibly associated with the very different temperatures and humidity in other regions that affect the infection, environmental survival, or epidemiology of different strains[36–39]. Furthermore, some clades managed to settle in other regions, possibly facilitated by super-spreading events such as the Mecca outbreak in 1987[27], and resulted in global pandemics (Fig. 7d). Finally, all recorded epidemics of *N. meningitidis*, except for the present W-135, seem to end in 10–20 years[36,40], and were gradually replaced by endemic lineages that have relatively stable frequencies over decades (Fig. 7e).

We proposed the model to reflect the exaggerated fluctuation of the population sizes of the epidemic lineages compared to the endemic ones, as well as its increasing international transmissions. A limitation is that the detailed evolutionary events, such as selections and recombinations, that resulted in these population dynamics, have not yet been incorporated into the model.

As shown above, the emergence of the current epidemic HC400_4 (ST-11) clade was not fully attributed to the decrease in MenA infections due to vaccines. The HC400_4 clade had capsular switching and started the population expansion in the 1960s, decades before the wide use of MenA vaccines. Some clusters in the HC400_4 clade also have started adapting to different geographic regions, resulting in global dissemination in the 1980s. Furthermore, although proposed in *N. meningitidis*, the model could also apply to many other pathogens. For example, *Vibrio cholerae* El Tor strains had undergone ~50 years of stepwise evolution and resulted in multiple local outbreaks before the onset of the 7th pandemic[41], and the *Salmonella enterica* ST313, which caused ~49,600 deaths every year, also had experienced >1000 years of stepwise evolution and population expansion[42]. Notably, not all lineages would survive through all four first stages. For example, HC300_31114 had been endemic in China since the 18th century, but had not expanded its population size. Moreover, the HC300_31213 clade managed to achieve stage b, and caused major disease outbreaks in China, but had never been established in any other region. However, both lineages would still potentially emerge if they went into higher stages in the model. Therefore, a genome-based disease surveillance system that characterizes endemic lineages by their current stages in the

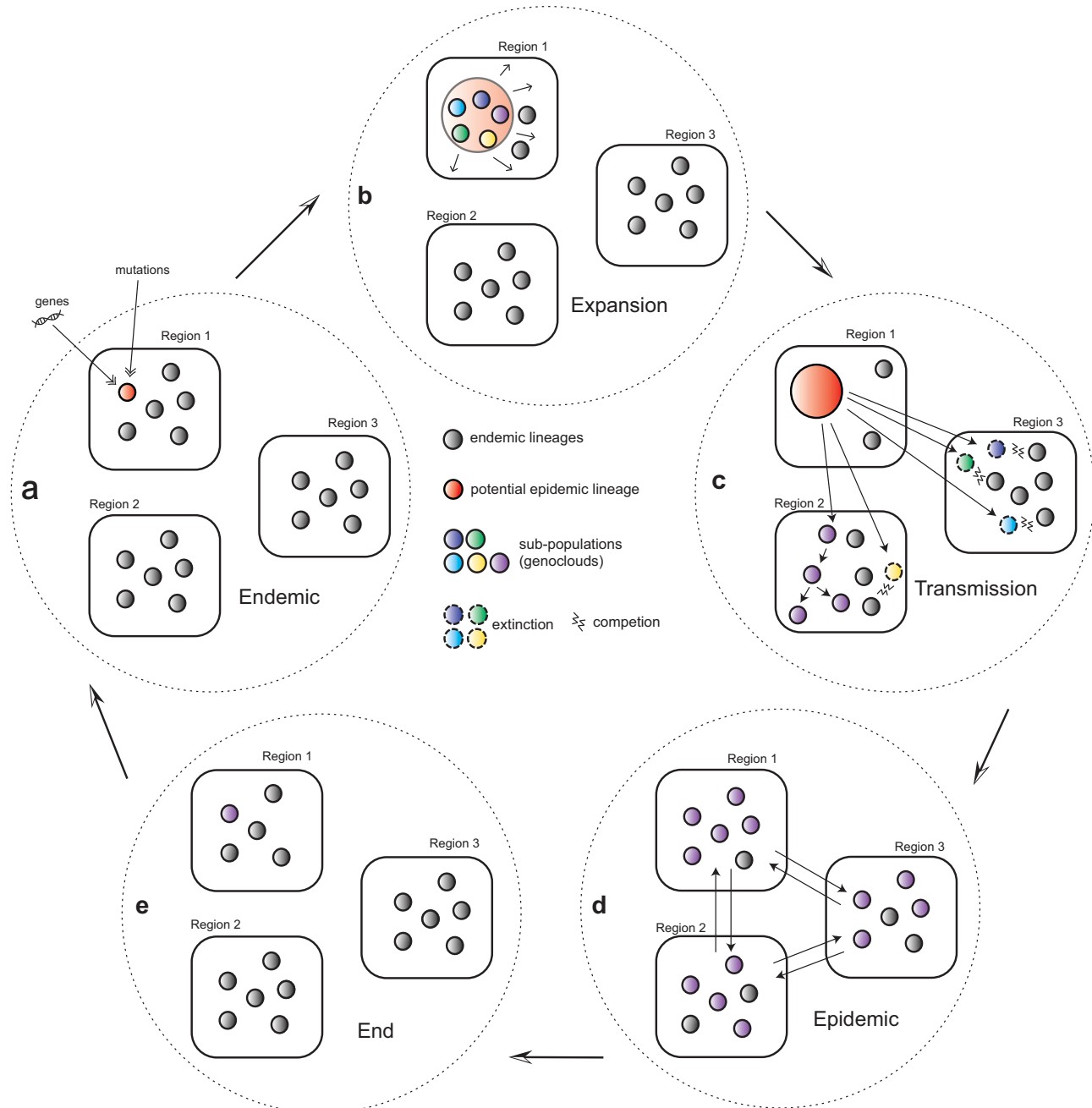

**Fig. 7 | Proposed model for epidemic dynamics in *N. meningitidis* and other bacterial pathogens. a** Certain local endemic lineage enhanced its virulence or transmissibility by gaining new genes or mutations, and (**b**) led to population expansion and diversification into multiple sub-populations. **c** Some of the subpopulations might be transmitted into other geographic regions. **d** Most of the transmissions did not last very long but some survived, resulting in epidemics or global pandemics. **e** The lineage could remain epidemic for several years, and finally be swept out in the long term by local lineages.

model will facilitate the recognition and control of new emerging epidemic lineages in the future.

We also employed HC10 for the identification of epidemiologically associated strains. *N. meningitidis* is an opportunistic pathogen that is also carried asymptomatically by healthy individuals. Traditional epidemiological investigations based on culture identification have high false positive rates[43]. HC10 clusters based on dcgMLST allowed rapid identification of genetically almost identical strains, and facilitated the epidemiological investigations in all five disease outbreaks in Suzhou. Similar approaches have been widely employed for the investigation of foodborne diseases due to *Salmonella* and *Escherichia*[44,45], demonstrating their high sensitivities and accuracies.

In summary, we present a distributed genotyping approach for *Neisseria* that does not depend on a central database. By integrating dcgMLST and HierCC, we revised species designations within the *Neisseria* genus and identified hundreds of intra-species populations, most of which have not been investigated before. With an overview of transmission patterns for epidemic lineages in *N. meningitidis* over the past 120 years, we proposed an epidemic model that separated the emergence, sustain, and end of epidemics into five stages. We also demonstrated the use of dcgMLST and HierCC in five disease outbreaks for the recognition of transmission chains. All these analyses demonstrated the significance of this system for the prevention, surveillance, and traceback of microbial pathogens in epidemics.

## Methods

### Ethics statement

This study was approved by the Institutional Review Board (IRB) of the Jiangsu Provincial Center for Disease Control and Prevention. The "Guidance of the Ministry of Science and Technology (MOST) for the Review and Approval of Human Genetic Resources" was not required for the study on human participants in accordance with the local legislation and institutional requirements. Written informed consent to participants in this study was provided by adult participants or the participants' legal guardians of child participants. No compensation was provided to the participants.

### Whole genome sequencing and data collection

The DNA of 149 isolates from Suzhou was purified and recovered by a silica gel column (D3146, HiPure Bacterial DNA kit) after incubation. Paired-end libraries with insert sizes of ~300 bp were prepared following Illumina standard genomic DNA library preparation procedure (VAHTS Universal DNA Library Prep kit for Illumina V3) and sequenced on an Illumina NovaSeq 6000 using the S4 reagent kits (v1.5) according to the manufacturer's instructions. The sequencing reads of each isolate were quality-trimmed and assembled into contigs using EToKi. In addition, 4411 assembled genomes were downloaded from the Gene-Bank database and 67,328 sets of short reads associated with *Neisseria* in the NCBI SRA were also assembled using the EToKi assemble pipeline[17]. To ensure the quality of the genomes, we kept only assemblies with N50 values between 0.2 MB and 1.5 MB, and total sizes between 1.9 and 4 MB were retained, resulting in a set of 69,845 genomes.

### Characterization of core genome and dcgMLST scheme

To identify core genes suitable for the dcgMLST scheme, we first built a set of representative genomes that retain most of the genetic diversities while removing redundancy caused by genetically virtually similar genomes. To do that, we assessed the pair-wise genetic distances of all genomes using Kssd[46], and grouped genomes into single-linkage clusters that shared ≥99.9% identity. One sequence with the greatest N50 value was chosen for each cluster, and was further quality checked by the presence of >37/40 single-copy core genes (SCGs) by fetchMGs[47]. For ease of biological interpretation of the analyses, we also included 29 additional complete genomes that have been widely studied, resulting in the final set of 7630 representative genomes annotated using Prokka[48].

Based on the representative genomes, a total of 13,984 pan genes were estimated using PEPPAN[49], and used as the basis for the whole-genome MLST scheme. Reference sequences for each pan gene were selected by choosing one allele for each cluster of ≥90% identities using EToKi MLSTdb. The MLSTdb module also identifies potential paralogous genes with ≥90% identities, which were removed from the scheme. We calculated the presence of pan genes in the representative genomes using the DTy pipeline, which is a modified, standalone version of EToKi MLSType[17] for dcgMLST schemes. Apart from the standalone databases, DTy used DIAMOND for amino acid-based searching of genes instead of Usearch in EToKi. Based on the results, we extracted a subset of 1149 core genes from them by selecting each gene that was (1) present in ≥95% of genomes, and (2) maintained intact open reading frames in >94% of its alleles. This resulted in a total of 1149 core genes, which were used as the basis of the distributed cgMLST scheme.

All genomes were then genotyped using the dcgMLST scheme by DTy again. Each allele in the dcgMLST was designated based on the MD5 hash value of its sequence, rather than an arbitrary sequential integer from a central database in the traditional cgMLST schemes. This allowed each genome to be characterized as a collection of up to 1149 MD5 hash values, which each represented a unique core gene allelic sequence. Similarly, the ST was also generated using the MD5 hash, based on a string consisting of MD5 hash values of the core genes. We then hierarchically grouped the allelic profiles of all genomes into multi-level clusters using pHierCC[20] in its development mode and statistically evaluated the consistencies and cohesiveness of each cluster using the pHCCeval module in the pHierCC package.

### Functional description of DTy

DTy is a generalized genotyping and clustering pipeline that consists of three modules: (1) Allele calling based on the distributed cgMLST scheme. The query genome (specified in --query) is compared with the reference alleles of the dcgMLST using both BLASTn and DIAMOND. The resulting allelic sequences are MD5-hashed into the 128-bit integers, which are used as allelic IDs. (2) The allelic profile of the query genome is compared with the allelic profiles of ~70,000 public genomes of *Neisseria*, and the closest neighbor with the minimum allelic difference was extracted. The HierCC assignments of the query genome are then updated based on the HierCC results of the closest neighbor. (3) Finally, species designation is predicted based on the mapping table between HC1050/HC1130 and *Neisseria* species.

### Prediction of serogroups, MLST types, and ANI95% clusters

In silico serotyping of *N. meningitidis* was performed using meningotype. The 7-gene MLST typing and CC designation were performed using BLASTn based on allelic sequences and profiles hosted in PubMLST[16]. FastANI[50] was employed to calculate the pairwise ANI of representative genomes, which were then grouped into single-linkage clusters with a threshold of 95%. The intra-species ANIs were calculated as ANIs between pairs of genomes within the same species. To estimate the inter-species ANI between neighboring species, we first calculated the average ANI value between each pair of species. Strains from two species that shared the greatest average ANI were then compared pairwisely using FastANI and the results were visualized as violin plots using ggplot2 package[51].

### Phylogenetic analysis

The maximum likelihood trees of the predominant HC760 groups were generated by the EToKi phylo module[17], had the recombinant regions removed using RecHMM[52], and visualized using GrapeTree[13]. The supertree was estimated using the cgMLSA package[19]. Briefly, cgMLSA built a gene tree for each core gene in the dcgMLST scheme. The gene trees were summarized together into a guide tree using ASTRID[53] and the posterior distribution of quartets was estimated using ASTRAL-III[54]. The guide tree was then split into many smaller disjoint subsets, each had its supertree estimated using ASTRAL-III. Finally, the supertree was built by concatenating all disjoint trees together and the branch lengths were estimated using ERaBLE[55].

### Bayesian inference of the epidemic HC lineages

The temporal origins of HC760_3, HC760_86, and HC760_4 lineages were estimated based on their maximum-likelihood trees using BactDating[56]. From the dataset, 499, 204, and 3975 genomes with clear collection times were selected for time origin analysis in each population, respectively. The phylodynamic inference of effective population size for time-scaled phylogenies was generated by skygrowth[57]. The geographic transmissions were estimated using TreeTime[58]. The temporal and spatial scale phylogenetic trees of the genomes were visualized using iTOL[59].

### Reporting summary

Further information on research design is available in the Nature Portfolio Reporting Summary linked to this article.

## Data availability

Source data are provided with this paper. The raw reads of 149 strains from this study have been deposited in the SRA under accession code

PRJNA1025302. The assembled genome sequences have been deposited in the Genebank database under accession code PRJNA1022841, and are also available in figshare at https://tinyurl.com/assembledNeisseria. The species tree of *Neisseria* is available at https://tinyurl.com/Neisseria-species. All accessions of genomes from the public database are included in Supplementary Data 1. The metadata supporting the conclusions of this article are also included in Supplementary Data 1. Source data for the figures (including supplementary figures) are provided in this paper. The reference sequences of core genes in the dcgMLST scheme, as well as the nomenclature results and HierCC profiles of all genomes, are provided as part of the DTy pipeline at https://doi.org/10.5281/zenodo.8396234. Source data are provided with this paper.

## Code availability

The code and detailed instruction manuals for running DTy can be found at https://doi.org/10.5281/zenodo.8396234.

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

## Acknowledgements

The project was supported by the National Natural Science Foundation of China (32170003, 32370099), the Jiangsu Specially-appointed Professor Project, and the Natural Science Foundation of Jiangsu Province (BK20211311). M.Z. was further supported by the Suzhou Science and Technology Innovations Project in Health Care (SKY2021013). L.Z. was supported by the Graduate Research and Innovation Projects of Jiangsu Province (KYCX22_3187). H.L. was supported by the National Natural Science Foundation of China (No. 82202465). L.S. was supported by the Suzhou Technology project (SS202079). Y.X. was further supported by the Research Program of Suzhou Key Technologies of Control and Prevention of Major Diseases and Infectious Diseases (GWZX202202).

## Author contributions

Conceptualization: Z.Z., L.Z., H.L., and M.Z.; resources: M.Z., Y.Z., B.W., H.Y., Q.S., Y.X., J.C., H.H., and B.P.; methodology: Z.Z., L.Z., H.L., Y.R., and M.Z.; software: Z.Z., L.Z., and H.L.; writing-original draft preparation: L.Z.; writing-review and editing: L.Z., M.Z., L.S., Y.Y., H.L., X.D., and Z.Z.; project administration: Z.Z.; All authors read and approved the final manuscript.

## Competing interests

The authors declare no competing interests.
