## [Peer Review File NEW · Nature Communications]

Distributed genotyping and clustering of *Neisseria* strains reveal continual emergence of epidemic meningococcus over a centuryReviewer #1 (Remarks to the Author):

In the manuscript "Distributed genotyping and clustering of *Neisseria* strains reveals continual emergence of epidemic meningococcus over a century" by Zhong and colleagues, the authors create a dataset with a global collection of genomes from the *Neisseria* species containing filtered sequencing data from nearly 70,000 isolates. They extract the core genome from 8,209 representative genomes and, following a sensible set of rules (gene present in >95% genomes and intact ORFs in >94% of alleles in the 70k dataset), select 1,149 core genes, which are used for the 'distributed cgMLST scheme'. Instead of assigning allele and sequence type (ST) numbers, the system works with MD5 hashes built from each unique sequence and STs. Then, they use hierarchical clustering of the dcgMLST with pHierCC. The resulting hierarchical clusters are deeply analyzed to evaluate species definition within the *Neisseria* genus, study persistence, evolutionary trajectories and transmission in *Neisseria meningitidis*. The authors also propose an epidemic model that includes the different evolutionary trajectories found on the main HCs.

It is a very interesting manuscript that untangles many points in the evolution of *Neisseria*. Overall, the paper is well written and very detailed, in fact, it needs a bit of simplification in my opinion, as it assumes the reader knows all the methodologies used. The methodology is quite well described and detailed for the construction of the dcgMLST scheme, phylogenetic analysis and phylodynamic inference and the algorithms used are appropriate. However, I was not familiar with pHierCC and missed some lines on i.e. I had to go to github to understand Silhouette scores.

Line 114: why does the name of hierarchical clusters is the same has the number of loci in the dcgMLST scheme (n=1,149)? This seems odd, may be a typo?

Figure 3: the authors should probably mark the two HC760 lineages on the tree, not only the HC300 subclusters.

Line 197: would be good to have in the text one sentence on how 'genoclouds' are defined.

Finally, the authors should show the confidence intervals on the inference of node ages on the two dated trees in figures 3 and 4, which is possible to obtain from BactDating.

Reviewer #2 (Remarks to the Author):

The manuscript is well written, clear, concise, and brings a lot of novelty on the population structure of *Neisseria* and the particular subtypes of *N meningitidis* having caused outbreaks or historical epidemics. I enjoyed reading it and the figures are very nice too.

I see three main themes in the manuscript: one concerns a novel approach or methodological development called distributed cgMLST; a second one deals with the phylogenetic structure of the *Neisseria* genus and population dynamics of epidemics of Nm; and a third is a model of Nm population biology / epidemics.

My overall impression is that the second theme was thoroughly covered and provides a great update and overview of Nm/genus population diversity, and to the contributions of genotyping to the definition of species and epidemic or endemic clones. This work is based on available genomic sequences and the author's own sequencing effort and provides a useful framework for future studies.

In contrast, the first theme is presented as a useful novelty, on which I fully agree, but there was little discussion of its significance, practicalities and possible shortcomings. Being highlighted in the title and introduction, I was expecting more information and discussion on this part. The third theme is a less strong part of the manuscript to my opinion; it presents a model but provides little justification in its favor (e.g., no positively selected mutations were described for the epidemic clones), and I was not convinced that it is a useful addition to this already considerable work.

dcgMLST and global strain tracking: This is a very interesting addition to the cgMLST approach and

one co-author (Z Zhou) is already known as of the leaders in cgMLST developments through his contributions to Enterobase and attached tools. dcgMLST is a very interesting concept as it enables comparing sequences without actually sharing nor disclosing them. Is this idea novel here? If not, please quote previous use/ideas. If yes, I think this innovation would deserve more attention/visibility than currently given in the paper.

While reading, a number of questions came to mind that were not addressed: how do we recognize the same clusters from a distinct study; Line 296: but does it require to re-analyse all genomes/sequences at the same time if the hashes are not made public in a repository or publication? Is there an incentive to publish the hashes of the STs or profiles of hashes so to enable forward comparisons? How does one compare MD5 hashes if there is no central repository for them? How does one create phylogenies from hashes if there is no reversibility into sequence data? Could the authors discuss the drawbacks of hashing alleles, e.g. how does one study recombination in sequences or phylogenies if these are not available? The Discussion starts with md5 hash system...but there was little focus on these questions.

Specific points

How was the production/development switch of HierCC chosen to create the HC groups?

How robust would the genus-wide cgMLST scheme be (paralogous loci removed? was the reproducibility of the typing scheme tested? what reference allele(s) is being used)?

Genoclouds and Genocloud clones: please define; can they be recognized by HC levels?

Line 401: There seems to be a disconnect between the use of dcgMLST for epidemiological understanding, and evolutionary model (positive selection not demonstrated, not linked to advantageous SNPs or other variation)

is ref 20 correct? Journal name Bioinformatics seems truncated

line 443: the ETOKi tool does not describe dcgMLST. could the authors point the reader to an open access tool to perform dcgMLST, perhaps integrating the hashing step of alleles and profiles, and their comparisons (could not find these in the github page provided)? I understand that the MLST package ETOKi can handle integers as well as hashes in exactly the same way, correct? If so it should be straightforward to use MD5 and ETOKi to implement this approach; but this could be made more explicit.

Line 490; it appears as an original solution to place assemblies in figShare; but why not having placed them in ENA or NCBI, where they would be searchable?

Line 494: remove "and the reference sequences of dcgMLST scheme loci"

Line 495-496: "The reference sequences of dcgMLST scheme loci are provided at <https://github.com/ADSGF203com/Neisseria-research-code>": I do not see the reference sequences there

Line 648: Ethical declarations: should be place here rather than begin of M&M ?

Figure 7: not sure this model is very novel or explanatory.

Fig 6: why are there core genes alignments needed; to extract sequence from assemblies? what are reference sequence, only one per locus; which one? how is the allele of the genome sequence extracted, is BLASTn used and how are the mismatches at extremities of hits (reference sequences) handled?

The abstract is in the past tense, unusual

Line 43 and elsewhere: 'resembled' might not be the best word; concordant with?

Is the HierCC system described in details (thresholds) and available (library of profiles) in some supplementary material? Will it be maintained by adding novel genomic sequences in the future?

Line 185: remove 'of'?

Line 242: notifiable has a special meaning in regulations on infectious diseases surveillance; it is the meaning here?

Line 259: extant lineages rather than nowadays?

Line 262: five

Line 284: rephrase 'strains that did not identify'

Line 290-291: a limitation of MLST or cgMLST as currently stands, is the need to share sequences, which may pose confidentiality issues. Otherwise there is no complexity I feel. Please be more precise in what dcgMLST brings compared to previous implementations

Line 299: 1000's integrers: are the hshes not 32 octets? Or did you mean, the STs? Not sure it is meaningful to compare STs without the hash alleles themselves, as most strains would have a distinct ST and you could not cluster them into the HC groups.

Lines 298-305: fine, but please discuss shortcomings of dcgMLST too (see above)

Lines 336-342: is the decline/frequency-dependent negative selection not due to immunity building against Nm serotypes? I doubt that salmonella and Klebsiella are good comparators here, and whether your model is generally applicable. This would require more studies I feel.

Line 356 'would expect': rephrase

Why use 'Darwinian' selection rather than positive selection? And clonal expansions could be the result of drift/neutral epidemic events rather than selection.

Line 364: 'possibly associated with their lack of adaption to environments in other regions' appears very speculative. Is there evidence?

Line 367: why are you using ref 35 on paratyphi as a reference here? I would have expected evidence for meningitidis instead

Line 368: also appears speculative to me " lineages that were relatively stable and better adapted"

Line 371: seroconversion: this typically has another meaning in medicine; correct here?

Line 374-385: the applicability of the model (which remains ill defined to my opinion) to other pathogens seems to me out of scope. I would suggest to remove the parts concerning the model; which do not seem strong nor necessary, really.

Figure 2 title: check

Fig 7: strains (not stains) – could be figure 1 (referred earlier)?

Responses to reviewers

-----Reviewer1-----

REVIEWER COMMENTS

Reviewer #1 (Remarks to the Author):

In the manuscript “Distributed genotyping and clustering of Neisseria strains reveals continual emergence of epidemic meningococcus over a century” by Zhong and colleagues, the authors create a dataset with a global collection of genomes from the Neisseria species containing filtered sequencing data from nearly 70,000 isolates. They extract the core genome from 8,209 representative genomes and, following a sensible set of rules (gene present in >95% genomes and intact ORFs in >94% of alleles in the 70k dataset), select 1,149 core genes, which are used for the ‘distributed cgMLST scheme’. Instead of assigning allele and sequence type (ST) numbers, the system works with MD5 hashes built from each unique sequence and STs. Then, they use hierarchical clustering of the dcgMLST with pHierCC. The resulting hierarchical clusters are deeply analyzed to evaluate species definition within the Neisseria genus, study persistence, evolutionary trajectories and transmission in Neisseria meningitidis. The authors also propose an epidemic model that includes the different evolutionary trajectories found on the main HCs.

It is a very interesting manuscript that untangles many points in the evolution of Neisseria. Overall, the paper is well written and very detailed, in fact, it needs a bit of simplification in my opinion, as it assumes the reader knows all the methodologies used. The methodology is quite well described and detailed for the construction of the dcgMLST scheme, phylogenetic analysis and phylodynamic inference and the algorithms used are appropriate. **However, I was not familiar with pHierCC and missed some lines on i.e. I had to go to github to understand Silhouette scores.**

- Thank you for the kind response. We have now added sentences to briefly describe what pHierCC did in the results section (see Q1 below), and also additional sentences in the methods section to describe the methodology and criteria we used while answering questions from both reviewers.

Q1.Line 114: Why does the name of hierarchical clusters is the same has the number of loci in the dcgMLST scheme (n=1,149)? This seems odd, may be a typo?

-We apologize for this unclear description. The hierarchical clusters were generated using cutoffs of 0, 1, 2, ...,1148 allelic differences. It is therefore the same number as the number of loci. We did not include HC1149, namely the 1149 allelic difference, because all genomes will fall into one group at that level. We have rewritten the paragraph (**now lines 116 to 119**) to read:

Furthermore, we assigned each of the dcgMLST allelic profiles using pHierCC into a set of 1,149 hierarchical clustering levels, from HC0, namely no allelic differences, to HC1148 where genomes that differed from each other by all but one alleles were grouped.

Q2.Figure 3: the authors should probably mark the two HC760 lineages on the tree, not only the HC300 subclusters.

-Thanks for the suggestion. Annotations for HC760 lineages have been added near the branch leading to the lineages, and an additional arc has been added to specify the associated strains in Figure 3 (**now Figure 4**).

Q3.Line 197: would be good to have in the text one sentence on how ‘genoclouds’ are defined.

-We apologize for this unclear description. ‘Genoclouds’ was initially proposed by Zhu et al. [PMID: 11287631]. They used both RFLP and gene-based sequence analysis to separate historical ST5/7 strains (HC760_74) into 9 genoclouds. We reused this definition to match our genomic data with these invaluable epidemiological records. We have now added an additional sentence to clarify the intention in **lines 202-207** that reads:

HC300_74 was generally consistent with STs 5 and 7 in legacy MLST and lineage III in MLEE. This clade was likely present in China before 1908 (CI95% 1900-1916) and later disseminated globally. A previous study subdivided strains in lineage III into nine so-called “genoclouds” based on Restriction Fragment Length Polymorphism (RFLP) and sequence analyses of six genes, tracking its international transmission for the first time. To compare our results with these historical records, we retrieved five of these genoclouds based on the genomic data (Fig.4a).

Q4.Finally, the authors should show the confidence intervals on the inference of node ages on the two dated trees in figures 3 and 4, which is possible to obtain from BactDating.

-Thank you for the advice. We have revised the world maps in Figures 3 and 4 (**now Figures 4 and 5**) to include the confidence intervals of major transmissions and also added confidence intervals to labeled branches in both figures.

-----**Reviewer2**-----

Reviewer #2 (Remarks to the Author):

The manuscript is well written, clear, concise, and brings a lot of novelty on the population structure of *Neisseria* and the particular subtypes of *N meningitidis* having caused outbreaks or historical epidemics. I enjoyed reading it and the figures are very nice too.

I see three main themes in the manuscript: one concerns a novel approach or methodological development called distributed cgMLST; a second one deals with the phylogenetic structure of the *Neisseria* genus and population dynamics of epidemics of Nm; and a third is a model of Nm population biology / epidemics.

My overall impression is that the second theme was thoroughly covered and provides a great update and overview of Nm/genus population diversity, and to the contributions of genotyping to the definition of species and epidemic or endemic clones. This work is based on available genomic sequences and the author's own sequencing effort and provides a useful framework for future studies.

In contrast, the first theme is presented as a useful novelty, on which I fully agree, but there was little discussion of its significance, practicalities and possible shortcomings. Being highlighted in the title and introduction, I was expecting more information and discussion on this part. The third theme is a less strong part of the manuscript to my opinion; it presents a model but provides little justification in its favor (e.g., no positively selected mutations were described for the epidemic clones), and I was not convinced that it is a useful addition to this already considerable work.

dcgMLST and global strain tracking: This is a very interesting addition to the cgMLST approach and one co-author (Z Zhou) is already known as one of the leaders in cgMLST developments through his contributions to Enterobase and attached tools. dcgMLST is a very interesting concept as it enables comparing sequences without actually sharing nor disclosing them. **Is this idea novel here? If not, please**

e quote previous use/ideas. If yes, I think this innovation would deserve more attention/visibility than currently given in the paper.

-Thanks for the comments. dcgMLST is a novel idea so far as I know. The MD5 hash itself is internally used in EnteroBase as part of the EToKi pipeline but has never been described anywhere, nor been used independently for allelic designations. We now add more paragraphs to describe this system in the manuscript (see Q5 & Q22)

While reading, a number of questions came to mind that were not addressed:

how do we recognize the same clusters from a distinct study;

-Thanks for the comments. HierCC provides a parameter “-a, --append” to add The NPZ output of a previous pHierCC run so that the same clusters from a distinct study can be recognized by using this parameter. Furthermore, we have now combined the HierCC with the nomenclature process into a new pipeline, DTy, which can be found at <https://github.com/ADSGF203com/DTy>. DTy also includes the dcgMLST+HierCC scheme for *Neisseria*, which can be directly used after installation.

Line 296: but does it require to re-analyse all genomes/sequences at the same time if the hashes are not made public in a repository or publication?

-Thanks very much for the comments. As mentioned above, the nomenclature results and HierCC profiles of all genomes in this study are now provided. New genomes now can be genotyped and compared with the ~70K genomes using DTy (see Q5 below).

Is there an incentive to publish the hashes of the STs or profiles of hashes so to enable forward comparisons?

-We appreciate the reviewer’s comments. The nomenclature results and HierCC profiles of all genomes are provided as part of the DTy pipeline at (see Q5):

<https://github.com/ADSGF203com/DTy/tree/master/db/Neisseria>

How does one compare MD5 hashes if there is no central repository for them?

-Thanks for the comments. Now the nomenclature results and HierCC profiles of all genomes, are provided as part of the DTy pipeline, and MD5 hashes can be compared by Neighbor-Joining(NJ) Algorithm which indicates phylogeny.

How does one create phylogenies from hashes if there is no reversibility into sequence data?

-Thanks very much. The MD5-based allelic profiles could be used to build minimum spanning trees or neighbor-joining trees using GrapeTree, which has been frequently used for cgMLST and implemented in both EnteroBase and pubmlst.org. However, it will indeed be complicated if one wants to do a sequence-based phylogeny of the alleles. We have included this potential shortcoming in the discussion (see Q22).

Could the authors discuss the drawbacks of hashing alleles, e.g. how does one study recombination in sequences or phylogenies if these are not available?

-Thank you for your kind suggestions. We add the paragraph to describe the limitations of dcgMLST (see Q22).

The Discussion starts with md5 hash system...but there was little focus on these questions.

-Thanks for the comments. We add paragraphs to describe the MD5 hash system in the Discussion (see Q5).

Q1.How was the production/development switch of HierCC chosen to create the HC groups?

-Thanks for the comments. The HierCC described here was generated solely in development mode. The production mode will be used for any new genomes after the publication (as implemented in the DTy repository in GitHub, see Q5).

We have rewritten the methods paragraph in **lines 475 – 478** to read:

We then hierarchically grouped the allelic profiles of all genomes into multi-level clusters using pHierCC in its development mode and statistically evaluated the consistencies and cohesiveness of each cluster using the pHCCeval module in the pHierCC package.

Q2. How robust would the genus-wide cgMLST scheme be (paralogue loci removed? was the reproducibility of the typing scheme tested? what reference allele(s) is being used)?

-We appreciate the reviewer's comments. The cgMLST scheme was developed as described and used the MLSTdb module in EToKi ¹. Paralogue loci that shared $\geq 90\%$ similarities to the others were removed. The resulting reference alleles were put in GitHub (see Q8). The dcgMLST scheme was initially built based on 7630 representative genomes, and we further evaluated its reproducibility by applying it to all 69,994 genomes. The results showed that all but three loci were still shared by $\geq 95\%$ of the genomes and we did not see any collision of the MD5 hash value, demonstrating its long-term sustainability. Furthermore, we have now included a stand-alone nomenclature pipeline for dcgMLST (DTy, see Q5), ensuring its reproducibility.

Reference

1. Zhou, Z. et al. The Enterobase user's guide, with case studies on Salmonella transmissions, Yersinia pestis phylogeny, and Escherichia core genomic diversity. Genome Res. 30, 138–152 (2020).

We have now modified the text to include some criteria for selecting core genes in the results (**lines 109-114**) that read:

We selected a set of 7630 representative sequences that encompassed most of the genetic diversity in the global collection and used them as the basis to establish a cgMLST scheme consisting of 1,149 core genes that were shared by $\geq 95\%$ of the representative genomes, following the same procedure outlined previously (see the Method for details). Furthermore, we applied the dcgMLST scheme to the global collection and found that 1,146 of the core genes were still shared by $\geq 95\%$ of the 69,994 strains, with no signal of collisions for the MD5 values.

Q3. Line 401: There seems to be a disconnect between the use of dcgMLST for epidemiological understanding, and evolutionary model (positive selection not demonstrated, not linked to advantageous SNPs or other variation)

-Thank you for the comments. The HC760 clusters generated by the dcgMLST scheme were used to evaluate the long-term persistence of *N. meningitidis* population on local/global scale. Yet, we do agree that the evolutionary model we proposed can not be used for evaluating sequence variations. It

is more of a descriptive model regarding the population dynamics (or say tree structures) of the epidemic lineage and could be evaluated based on the international transmission frequencies and fluctuations of population sizes. We added a paragraph in **lines 388 to 391** to discuss the limitation of the model, which reads:

We proposed the model to reflect the exaggerated fluctuation of the population sizes of the epidemic lineages compared to the endemic ones, as well as its increasing international transmissions. A limitation is that the detailed evolutionary events, such as selections and recombinations, that resulted in these population dynamics, have not yet been incorporated into the model.

Q4.Is ref 20 correct? Journal name Bionformatics seems truncated

-Corrected. Many thanks.

Q5.line 443: the EToKi tool does not describe dcgMLST. could the authors point the reader to an open access tool to perform dcgMLST, perhaps integrating the hashing step of alleles and profiles, and their comparisons (could not find these in the github page provided)? I understand that the MLST package ETOki can handle integers as well as hashes in exactly the same way, correct? If so it should be straightforward to use MD5 and EToKi to implement this approach; but this could be made more explicit.

-We appreciate the reviewer's comments. We agree with the reviewer that the use of EToKi for dcgMLST has not been well documented on its GitHub page. In order to facilitate the use of the new scheme, we extracted the nomenclature module of EToKi and made it an independent pipeline as DTy (<https://github.com/ADSGF203com/DTy>). Another change is that we now use BLASTn+DIAMOND instead of BLASTn+Usearch for allele searches because Usearch is not open source. Furthermore, this repository also includes all required files for the dcgMLST+HierCC scheme in *Neisseria*, including the reference sequences of all core genes, the allelic profiles, and the HierCC results for ~70000 genomes. We have now added sentences describing the pipeline in the **Results (lines 125-129)**:

To facilitate the application of the dcgMLST scheme, we implemented an automatic pipeline, DTy (<https://github.com/ADSGF203com/DTy>), which automatically genotypes a *Neisseria* genome and assigns it into multi-level HC clusters and species after comparing them with the representative genomes.

in **Discussion (lines 310-314)**:

For example, the allelic profiles of ~70,000 *Neisseria* genomes occupied 67.2MB of storage space and could be easily shared together with the DTy pipeline in public storage spaces. As a comparison, the *Clostridioides* cgMLST in EnteroBase currently hosts 29,085 genomes, ~40% as large as the *Neisseria* genomes here, and has registered 463K alleles that account for ~600MB of storage spaces.

and in **Methods (lines 459-468)**:

Reference sequences for each pan gene were selected by choosing one allele for each cluster of $\geq 90\%$ identities using EToKi MLSTdb. The MLSTdb module also identifies potential paralogous genes with $\geq 90\%$ identities, which were removed from the scheme. We calculated the presence of pan genes in the representative genomes using the DTy pipeline, which is a modified, standalone version of EToKi MLSType¹ for dcgMLST schemes. Apart from the standalone databases, DTy used DIAMOND for amino acid-based searching of genes instead of Usearch in EToKi. Based on the results, we extracted a subset of 1149 core genes from them by selecting each gene that was (1) present in $\geq 95\%$ of genomes, and (2) maintained intact open reading frames in $>94\%$ of its alleles. This resulted in a total of 1,149 core genes, which were used as the basis of the distributed cgMLST scheme.

and a paragraph describing the function of DTy in **lines 480-488**:

Functional description of DTy

DTy is a generalized genotyping and clustering pipeline that consists of three modules: (1) Allele calling based on distributed cgMLST schemes. The query genome (specified in --query) is compared with the reference alleles of the dcgMLST using both BLASTn and DIAMOND. The resulting allelic sequences were MD5 hashed the 128-bit integer is used as allelic IDs. (2) The allelic profile of the query genome is compared with the allelic profiles of ~70,000 public genomes of *Neisseria*, and the closest neighbor with the minimum allelic difference was extracted. The HierCC assignments of the query genome are then updated based on the HierCC results of the closest neighbor. (3) The query genome is then assigned a species prediction based on the mapping table between HC1050/HC1130 and *Neisseria* species.

A code availability section (**lines 534 to 536**) was also added that reads:

Code availability

The code and detailed instruction manuals for running DTy can be found at <https://github.com/ADSGF203com/DTy>.

Q6.Line 490:it appears as an original solution to place assemblies in figShare:

but why not having placed them in ENA or NCBI, where they would be searchable?

-Thanks very much for the comments. Now the assembled genome sequences have been deposited in the Genome Warehouse (GWH) in the National Genomics Data Center with Bioproject accession **PRJCA018226**.

Q7.Line 494: remove "and the reference sequences of dcgMLST scheme loci".

- Corrected.

Q8.Line 495-496: “The reference sequences of dcgMLST scheme loci are provided at <https://github.com/ADSGF203com/Neisseria-research-code>; I do not see the reference sequences there.

-We apologize for this mistake. Reference sequences of alleles have been integrated as part of the DTy pipeline and hosted in

<https://github.com/ADSGF203com/DTy/blob/master/db/Neisseria/references.fas.gz>

The data availability section has also been changed accordingly to read:

Lines 529-532:

The reference sequences of core genes in the dcgMLST scheme, as well as the nomenclature results and HierCC profiles of all genomes, are provided as part of the DTy pipeline at <https://github.com/ADSGF203com/DTy/tree/master/db/Neisseria>.

Q9.Line 648: Ethical declarations: should be place here rather than begin of M&M ?

- Thanks for the comments. We read several articles

(<https://www.nature.com/articles/s41467-023-40718-8>,

<https://www.nature.com/articles/s41467-023-40706-y>) recently published in Nature Communications

and found that the Ethical declarations are always placed after Author Information. So we kept the

paragraph there but are happy to move it if needed. There is already a section of the “Ethics

statement” regarding the involved patient samples at the beginning of M&M.

Q10.Figure 7: not sure this model is very novel or explanatory.

- Thank you for the comments. We described the intention and novelty of this model in answers to Q3, Q23, and Q26 separately.

Q11.Fig 6: why are there core genes alignments needed; to extract sequence from assemblies? what are reference sequence, only one per locus; which one? how is the allele of the genome sequence extracted, is BLASTn used and how are the mismatches at extremities of hits (reference sequences) handled?

- Thank you for all the comments. The “core genes alignments” is more of an inappropriate choice of words. We meant to, and have modified the words to “similarity-based search of core genes”. The reference sequences of core genes were picked using the MLSTdb module in the EToKi package, which selects alleles of <90% similarity for each core gene. The resulting reference sequences were stored together with DTy (see Q5). Both BLASTn and DIAMOND were used for core gene alignments and an ‘End-to-End’ alignment is required for calling an allele. The methods section has also been modified accordingly (see Q5).

Q12.The abstract is in the past tense, unusual.

-Corrected.

Q13.Line 43 and elsewhere: ‘resembled’ might not be the best word; concordant with?

-Corrected.

Q14.Is the HierCC system described in details (thresholds) and available (library of profiles) in some supplementary material? Will it be maintained by adding novel genomic sequences in the future?

- We appreciate the reviewer’s comments. The HierCC stores all possible single-linkage clusters from allelic differences of 0 to 1148. We have now included all the allelic profiles and HierCC results in the DTy pipeline at <https://github.com/ADSGF203com/DTy/tree/master/db/Neisseria>, which will be regularly updated.

Q15.Line 185: remove ‘of’ ?

-Corrected.

Q16.Line 242: notifiable has a special meaning in regulations on infectious diseases surveillance; it is the meaning here?

-Thanks. Changed to “recognized clone”

Q17.Line 259: extant lineages rather than nowadays?

-Corrected.

Q18.Line 262: five

-Corrected.

Q19.Line 284: rephrase ‘strains that did not identify’

-Thanks. Changed to “strains that haven’t been characterized in the culture.”

Q20.Line 290-291: a limitation of MLST or cgMLST as currently stands, is the need to share sequences, which may pose confidentiality issues. Otherwise, there is no complexity I feel. Please be more precise in what dcgMLST brings compared to previous implementations.

- Thanks for the comments. Besides the confidentiality, I also found that the allelic database is approximately >10X as large as the profiles. Furthermore, any new allele that gets into the central allelic databases needs to be compared with all existing ones. This operation sounds trivial but grows linearly with the number of alleles. I benchmarked in Enterobase 2 years ago and a maximum throughput of ~100 new alleles per second is expected when there are > 1 million records. This is currently fine but will be exaggerated in the future when the database gets larger and grows faster.

We have added sentences in the discussion that read:

Lines 310-315:

As a comparison, the *Clostridioides* cgMLST in EnteroBase currently hosts 29,085 genomes, ~40% as large as the *Neisseria* genomes here, and has registered 463K alleles that account for ~600MB of storage spaces. The dcgMLST also replaced the allelic comparison process, which scales linearly with the size of the database, to MD5 hash of constant time complexity.

Q21.Line 299: 1000s integrers: are the hshes not 32 octets? Or did you mean, the STs? Not sure it is meaningful to compare STs without the hash alleles themselves, as most strains would hav e a distinct ST and you could not cluster them into the HC groups.

-We sincerely value the reviewers' feedback. The hashes are 128-bit integers or 32-octets, which are the same. The resulting space usage is estimated to be 67.2 MB for ~70000 genomes, or ~960 bytes per genome after compression. And yes, we compare STs based on their associated allelic profiles, which are 1000's of hash values, and the HC groups were also assigned based on them.

Q22.Lines 298-305: fine, but please discuss shortcomings of dcgMLST too (see above)

-Thank you for your kind suggestions. We add the following paragraph to describe the limitations of dcgMLST as follows. **Please check the new lines 322 to 326 in the manuscript.**

There are, of course, some limitations for dcgMLST. Multi-locus sequence analysis (MLSA), a detailed phylogenetic analysis based on the sequences of genes in the MLST schemes³¹, will hardly apply to records in the dcgMLST scheme due to a lack of sequence information. Also, while the allelic designations have been de-centralized, there is still a need for public databases for sharing allelic profiles and metadata of bacterial strains for biological interpretation of the results.

Q23.Lines 336-342: is the decline/frequency-dependent negative selection not due to immunity building against Nm serotypes? I doubt that *Salmonella* and *Klebsiella* are good comparators here, and whether your model is generally applicable. This would require more studies I feel.

- We greatly value the reviewer's feedback. You proposed an interesting assumption of the immunity-based selection. However, a more intriguing question is why the non-epidemic lineages in *Neisseria* can survive for decades, or even longer, without a clear signal of population switch. And this is one of the supports for our proposed endemic-epidemic model. While the epidemic lineages fluctuated quickly, due to whatever selection they faced, the non-epidemic lineages, while also

primarily found in humans, have not been selected at the same speed. Moreover, these non-epidemic (or endemic) lineages were often geographically restricted and unlikely to be found across multiple countries. I agree that our model is more of a descriptive one, and might not be applied to other bacteria, but I think the phenomenon is interesting enough to be kept in the manuscript. Meanwhile, we agree with the reviewer and have removed the *Salmonella* and *Klebsiella* parts, as mentioned. Thanks again.

Q24.Line 356 ‘would expect’: rephrase.

-Corrected.

Q25.Why use ‘Darwinian’ selection rather than positive selection? And clonal expansions could be the result of drift/neutral epidemic events rather than selection.

-Thanks very much. You are correct that the “Darwinian selection” and the “positive selection” point to about the same thing. We kept the word “Transient Darwinian selection” in line 368 because this is the name of the model we used nine years ago¹, and changed the word “Darwinian selection” in line 374 to “positive selection”. We also added the possibility of genetic drift there.

The sentence in **lines 373-374** now reads: “Some endemic lineages would tend to expand their population sizes due to short-term positive selection or by genetic drift (Fig. 7b).”

Reference

1. Zhou, Z. et al. Transient Darwinian selection in *Salmonella enterica* serovar Paratyphi A during 450 years of global spread of enteric fever. *Proc. Natl. Acad. Sci. U. S. A.* 111, 12199–12204 (2014).

Q26.Line 364: ‘possibly associated with their lack of adaption to environments in other regions’ appears very speculative. Is there evidence?

- We greatly value the reviewer's feedback. Unfortunately, there is no direct experimental evidence for this. However, previous studies showed that the abilities of infection¹ and environmental survival² for *N. meningitidis* vary with strains. And epidemic patterns of *N. meningitidis* infections varied with countries³, even in different parts of the meningitidis belt⁴. We also found that there

are >10 recorded transmissions of MenA or MenW135 strains (see Figures 4 and 5) into other countries before their global epidemics, but none survived very long. We agree that the current description is too speculative, and revised the sentences to read:

Now lines 379 to 381

However, most of these early transmissions would not lead to long-lasting disease outbreaks, possibly associated with the very different temperatures and humidity in other regions that affect the infection, environmental survival, or epidemiology of different strains³⁵⁻³⁸.

Reference

1. Loh, E. *et al.* Temperature triggers immune evasion by *Neisseria meningitidis*. *Nature* **502**, 237–240 (2013).
2. Swain, C. L., Martin, D. R., Sim, D., Jordan, T. W. & Mackichan, J. K. Survival of *Neisseria meningitidis* outside of the host: environmental effects and differences among strains. *Epidemiol. Infect.* **145**, 3525–3534 (2017).
3. Peltola, H. Meningococcal disease: still with us. *Rev. Infect. Dis.* **5**, 71–91 (1983).
4. Pinilla-Monsalve, G. D. *et al.* Socioepidemiological macro-determinants associated with the cumulative incidence of bacterial meningitis: A focus on the African Meningitis Belt. *Front. Neurol.* **14**, 1088182 (2023).

Q27.Line 367: why are you using ref 35 on paratyphi as a reference here? I would have expected evidence for meningitidis instead.

- Thank you for the comment. We decided to describe the phenomenon without giving further assumptions. The sentence now reads:

Now lines 383 to 386

Finally, all recorded epidemics of *N. meningitidis*, except for the present W-135, seem to end in 10-20 years^{35,39}, and were gradually replaced by endemic lineages that have relatively stable frequencies over decades (Fig. 7e)

Q28.Line 368: also appears speculative to me “ lineages that were relatively stable and better adapted”

- Thank you for pointing this out. We see the long-term persistence of local, endemic lineages based on the genomic data, but have not got enough evidence for their better adaptation. So the sentence has now been rewritten. Please check Q27 above.

Q29.Line 371: seroconversion: this typically has another meaning in medicine; correct here?

- We have changed the word to “capsular switching”. Thanks.

Q30.Line 374-385: the applicability of the model (which remains ill defined to my opinion) to other pathogens seems to me out of scope. I would suggest to remove the parts concerning the model; which do not seem strong nor necessary, really.

-Thanks for the suggestions. We have removed this part.

Q31.Figure 2 title: check

-Corrected.

Q32.Fig 7: strains (not stains) – could be figure 1 (referred earlier)?

-Thank you for your kind suggestions. We have now renamed it as Figure 1, and referred to it in line 116 which reads:

Now lines 114-116

In particular, we invented an algorithm that designated each core gene allele as the MD5 hash value of the sequence of the gene, and thus enabled distributed cgMLST (dcgMLST) nomenclature without a central database (Fig. 1)

Reviewer #1 (Remarks to the Author):

The authors have properly responded to my suggestions and comments. I appreciate this and approve this version for publication.

Reviewer #2 (Remarks to the Author):

The authors have addressed my previous comments completely and adequately. I thank them for taking them into account and congratulate them for implementing the DTy pipeline. Best wishes,
Sylvain Brisse